# Unplanned nursing home admission among discharged polymedicated older inpatients: a single-centre, registry-based study in Switzerland

Filipa Pereira ![ORCID] ,[1,2] Henk Verloo ![ORCID] ,[2,3] Armin von Gunten,[3] María del Río Carral,[4] Carla Meyer-Massetti,[5,6] Maria Manuela Martins,[7] Boris Wernli[8]

For numbered affiliations see end of article.

**Correspondence to**
Filipa Pereira;
filipa.pereira@hevs.ch

## ABSTRACT

**Objective** To investigate patient characteristics and the available health and drug data associated with unplanned nursing home admission following an acute hospital admission or readmission.

**Design** A population-based hospital registry study.

**Setting** A public hospital in southern Switzerland (Valais Hospital).

**Participants** We explored a population-based longitudinal dataset of 14 705 hospital admissions from 2015 to 2018.

**Outcome measures** Sociodemographic, health and drug data, and their interactions predicting the risk of unplanned nursing home admission.

**Results** The mean prevalence of unplanned nursing home admission after hospital discharge was 6.1% (n=903/N=14 705). Our predictive analysis revealed that the oldest adults (OR=1.07 for each additional year of age; 95% CI 1.05 to 1.08) presenting with impaired functional mobility (OR=3.22; 95% CI 2.67 to 3.87), dependency in the activities of daily living (OR=4.62; 95% CI 3.76 to 5.67), cognitive impairment (OR=3.75; 95% CI 3.06 to 4.59) and traumatic injuries (OR=1.58; 95% CI 1.25 to 2.01) had a higher probability of unplanned nursing home admission. The number of International Classification of Diseases, 10th version diagnoses had no significant impact on nursing home admissions, contrarily to the number of prescribed drugs (OR=1.17; 95% CI 1.15 to 1.19). Antiemetics/antinauseants (OR=2.53; 95% CI 1.21 to 5.30), digestives (OR=1.78; 95% CI 1.09 to 2.90), psycholeptics (OR=1.76; 95% CI 1.60 to 1.93), antiepileptics (OR=1.49; 95% CI 1.25 to 1.79) and anti-Parkinson's drugs (OR=1.40; 95% CI 1.12 to 1.75) were strongly linked to unplanned nursing home admission.

**Conclusions** Numerous risk factors for unplanned nursing home admission were identified. To prevent the adverse health outcomes that precipitate acute hospitalisations and unplanned nursing home admissions, ambulatory care providers should consider these risk factors in their care planning for older adults before they reach a state requiring hospitalisation.

## INTRODUCTION

The hospitalisation of home-dwelling older adults, for any reason and even for a short admission, can lead to substantial functional

### Strengths and limitations of this study

► A hospital registry of 14 705 hospital admissions, involving 9430 different polymedicated older adults admitted from their homes, was analysed to determine the risk of unplanned nursing home admission.

► Bivariate analyses were conducted on independent variables, and generalised estimating equations were computed to predict how sets of predictors influenced the adjusted probability of unplanned nursing home admission.

► Causality analysis was not feasible based on the nature of the routinely collected data.

► Although the study considered statistical associations between drugs and unplanned nursing home admission, it did not use clinically diagnosed drug–drug interactions.

► Our data were unable to identify hospitalisations that might have been triggered by limited home care options or those that became necessary while older adults awaited a place in a nursing home.

decline.[1 2] Both their health disorder itself and the hospital environment can foster such functional decline, increase the risk of future illness and irreversibly diminish their quality of life.[1 2] Most hospitalised older adult inpatients wish to return home and continue their everyday life as before. However, these different factors may hinder this wish at discharge.[3 4] The unmet patient needs related to functional decline and safety after returning home can lead to a higher risk of hospital and emergency department readmissions and thus to subsequent unplanned nursing home admission.[5] After hospitalisation, an unplanned nursing home admission can be a devastating and overwhelming experience for older adults and their relatives, and it increases overall healthcare system costs.[6]

Whether planned or unplanned, nursing home admission commonly follows two paths: (1) within the community, directly

from home, or (2) from hospital, directly transitioning from hospital discharge.[2] In the community, transitions to nursing homes are generally the result of thoughtful decisions made by home-dwelling older adults, their families, and healthcare and social care providers based on the evolution of the person's long-term health and functional state or on an acute decline and corresponding increase in care needs that cannot be met at home. Recent findings have suggested that the predictors of nursing home admission are mainly based on underlying cognitive and functional impairments combined with a lack of support and assistance in daily living at home.[7]

The causes of unplanned nursing home admission directly after acute hospital discharge are heterogeneous. There are several reasons why older adults may require long-term care—that cannot be provided in a community setting—following acute hospital admission, for example, a new medical problem or the worsening of existing chronic disease(s) entailing dependency and requiring complex forms of care. Furthermore, there may be a breakdown of family circumstances and/or lack of social support.

Bellelli *et al* showed that advanced age (OR=4.8; 95% CI 2.6 to 8.9, p<0.001), cognitive impairment (OR=2.3; 95% CI 1.4 to 3.9, p<0.001) and poor functional status (OR=10.2; 95% CI 4.7 to 22.5, p<001) at discharge from a rehabilitation unit were the main predictors of subsequent nursing home admission.[8] The integrative review by Fogg *et al* found a similar result for cognitive impairment (OR=2.14; 95% CI 1.24 to 3.70, p<0.001).[9] A randomised controlled trial by Landefeld *et al* found that older inpatients in an acute care medical unit with a decline in their ability to perform one or more of the basic activities of daily living (ADLs) were more often discharged to a nursing home than those with less functional decline (22% and 14%, respectively; p<0.01).[10] Ferrucci *et al* identified stroke, cancer, congestive heart failure, pneumonia, coronary heart disease and hip fractures as the leading medical precipitators of functional decline and nursing home admission.[11] Older adult inpatients are frequently subject to iatrogenic events during hospitalisation, including adverse drug reactions, nosocomial infections, and the consequences of falls, fractures, and using chemical or physical restraints.[12] Such events can lengthen hospitalisation, produce cognitive changes and lessen the ability to perform the ADL, all potentially leading to unplanned nursing home admission.[12] Indeed, hospitalisation causes an increased risk of the onset of acute cognitive decline in the form of delirium, with a prevalence of up to 60% on some surgical wards,[13] often leading to unplanned nursing home admission.[14] Dementia, Parkinson's disease and its associated risk of falls, and behavioural changes are common reasons for deciding to transfer inpatients from hospital to long-term care.[15 16]

Polypharmacy has been associated with adverse health outcomes among home-dwelling older adults.[17] Some prospective studies with small samples have established relationships between drug treatments during acute hospitalisation and unplanned nursing home admission.[18] Cardiovascular drugs (particularly vasodilators, diuretics and anticoagulants), drugs against diabetes, steroids, non-steroidal anti-inflammatory drugs, opiates, antibiotics, anticholinergics and benzodiazepines have all been associated with unplanned nursing home admission.[18]

To the best of our knowledge, and despite more frequent post-discharge nursing home admissions in Switzerland than in other countries, there is scarce research exploring how unplanned admissions to nursing homes are related to prior hospitalisation.[19] The present study aimed to investigate the associations between polymedicated older inpatients' sociodemographic and clinical characteristics, drug data and their interactions, and their unplanned nursing home admission following an acute care hospital stay.

## METHODS
### Study design
The present population-based hospital registry study was conducted with close regard to the REporting of studies Conducted using Observational Routinely collected health Data statement.

### Population and data collection
Our 4-year, longitudinal, population-based hospital registry of electronic health records included polymedicated (five and more drugs prescribed) home-dwelling older adults admitted and readmitted to the Valais Hospital, a multisite public teaching hospital (1074 beds) in southern Switzerland with a mean annual number of hospitalisations of approximately 39 000. This registry continues to be analysed as part of a larger project.[20] Our study defined 'unplanned nursing home admission' as the impossibility for a formerly home-dwelling older adult inpatient to return there after hospital discharge, and this included any new admission to a nursing home following an acute care admission.[2] All the patients included in the study followed a home to hospital to nursing home pathway. Nursing homes do not expect their residents to return to independent living in the community. The extracted patient data contained sociodemographic characteristics, medical and surgical diagnoses, routinely assessed clinical data (such as gait, falls risk, hearing or pain) and the drugs prescribed. The medical and surgical diagnoses encoded diagnostic data using the WHO International Classification of Diseases, 10th version (ICD-10), and the Swiss Classification of Surgical Interventions (CHOP).[21] The hospital dataset showed that discharged patients had been prescribed 2370 different medicines. The Anatomical Therapeutic Chemical (ATC) classification system's 14 top-level codes were used to structure that dataset of prescribed medicines.[22] The extracted data, from multiple dataset sources, were transformed and synthesised using best practices.[23] Our dataset was composed of 14 705 hospital admissions from home settings between 2015 and 2018. Data were without missing values, and there were similar numbers

of annual hospital admissions: 3777, 3534, 3724 and 3670, respectively.

## Patient and public involvement

Patients were not involved in the development of the research questions, study design, outcome measures or the conduct of the study.

## Dataset customising for predictive analysis
### Synthesising the extracted data

Since where patients had arrived from and where they were discharged to were two distinct variables, the dataset was recoded and customised to identify the number of older adult inpatients admitted straight from their homes and then discharged to a nursing home (n=903) or returning to their homes (n=13 802), as presented in a previous paper.[24] Therefore, older adults who died during hospitalisation (as assessed by the Valais Hospital's healthcare staff) were automatically excluded (n=131). Each subject's unique identifier was used to distinguish between different observations from 2015 to 2018 and to account for hospital readmissions. Cases involved 9430 different older adults, with an average of 1.56 hospital stays per person. Sociodemographic and clinical data were considered independent variables and used to compute the predictive models.[24] Unplanned nursing home admission after discharge from our participating hospital between 2015 and 2018 was identified by the difference between the original abode (home) and the destination at discharge (a nursing home or their own home), and this was used as the dependent variable of interest.

### Sociodemographic and hospital variables

The analysis included two sociodemographic control variables: age and sex. Age was considered a continuous variable; its progressive impact was conclusive in preliminary investigations and previous studies.[25]

### Health variables

Numerous variables were used to describe older adults' health status at the end of their hospital stay. The modelling analysis included three of the six hierarchical clusters preliminarily computed as being variables significantly associated with more unplanned nursing home admissions in the descriptive analysis: mobility, dependency in the ADL and cognitive status.[24] Cognitive status was measured at an ordinal level using five categorical variables (perception–alertness, orientation, attention, decision-making process and ability to learn). Finally, the year of hospitalisation was introduced as a control variable, based on the fact that hospital admissions occurring earlier in the 4-year study were associated with a higher probability of unplanned nursing home admission.[24]

### Drugs

The WHO ATC classification system[22] was used to select frequently prescribed drugs at discharge as independent variables for the predictive model. The selection of drug class interactions was based on a literature review and expert opinions.[26] A cut-off point of at least 30 subjects per drug category prescribed was necessary to have a critical mass of data for computing robust statistics. The number of drugs prescribed at hospital discharge was considered continuous.

## Data analysis strategy

Data were extracted into a Microsoft Excel spreadsheet (Microsoft, Redmond, Washington, USA) and subsequently imported into SPSS software, V.26.0 (IBM Corp). Associations with unplanned nursing home admission were examined based on previous studies: patient age and sex, hospital length of stay, the principal and secondary ICD-10 diagnoses, surgical interventions (CHOP) and prescribed drugs. No causality analyses were possible because data analysis was retrospective and based on routine data: there was no way of knowing medication regimens or functional status before hospitalisation and how these might be associated with unplanned nursing home admission. A series of unadjusted bivariate analyses using cross-tabulations were conducted to investigate whether the sociodemographic, health and drugs data (more than one independent variable) were statistically significantly associated with unplanned nursing home admission from 2015 to 2018 (our single dichotomous outcome). In a second stage, a series of generalised estimating equations (GEE or population-averaged logistic regression models) were computed to predict how sets of predictors influenced the probability of unplanned nursing home admission. The variables entered at the first stage were derived from the significant associations between sociodemographic characteristics, clinical and medical conditions and unplanned nursing home admission (table 1). The multivariable analysis model included 52 level 2 ATC drug classes, respecting the good practices for logistical regressions involving large population-based samples.[27] This adjusted baseline model was then completed by adding drugs that were found to be significantly associated with unplanned nursing home admissions in the previous analysis. Lastly, based on our literature review, known drug–drug interactions between different ATC drug classes were added to the baseline model. The model estimated each predictor's impact, other things being equal, by estimating its net impact controlling for confounding factors (adjusted ORs). Since the data are based on a whole population, not a sample, the ORs' CIs and statistical tests were used to indicate the robustness of relationships since they normally only make sense for statistical inference.

## RESULTS
### Population description

Fifty-five per cent of the population sample were men, and the total sample's mean age was 78.16 years old (SD=7.65). Mean hospital length of stay was 8.63 days (SD=7.58). The mean number of drugs prescribed at hospital discharge

**Table 1** Prevalence of unplanned nursing home admissions with regard to associations with sociodemographic characteristics and clinical and medical conditions among polymedicated hospitalised older adults (N=14 705)

| Variables | Unplanned nursing home admission, n (%) | P value |
|---|---|---|
| Overall sample of older adults (n=14 705) | 903 (6.1) | |
| Sex | | <0.001 |
| Female/male | 575 (8.8)/328 (4.0) | |
| Age in years | | <0.001 |
| 65–69 | 49 (2.2) | |
| 70–79 | 192 (3.2) | |
| 80–89 | 437 (8.3) | |
| 90 or more | 225 (19.7) | |
| Mobility | | <0.001 |
| Full ability (0)/impairment (1) | 214 (2.0)/689 (16.7) | |
| Dependence in the activities of daily living | | <0.001 |
| Full ability (0)/impairment (1) | 472 (3.4)/431 (44.8) | |
| Mental status | | <0.001 |
| Full ability (0)/impairment (1) | 531 (3.8)/372 (41.3) | |
| ICD-10 principal diagnosis: circulatory problems | | <0.001 |
| No (0)/yes (1) | 752 (6.7)/151 (4.3) | |
| ICD-10 principal diagnosis: infection | | 0.003 |
| No (0)/yes (1) | 892 (6.2)/11 (2.7) | |
| ICD-10 principal diagnosis: respiratory problems | | 0.226 |
| No (0)/yes (1) | 797 (6.1)/106 (6.8) | |
| ICD-10 principal diagnosis: traumatic injuries | | <0.001 |
| No (0)/yes (1) | 720 (5.3)/183 (14.9) | |
| ICD-10 principal diagnosis: tumour | | 0.001 |
| No (0)/yes (1) | 835 (6.4)/68 (4.3) | |
| Number of ICD-10 diseases | | <0.001 |
| 1 | 5 (1.8) | |
| 2 | 17 (2.9) | |
| 3 | 37 (3.9) | |
| 4 | 47 (3.9) | |
| 5 or more | 797 (6.8) | |
| Number of surgical interventions (CHOP) | | <0.001 |
| 0 | 379 (7.8) | |
| 1 | 187 (6.4) | |
| 2 | 135 (5.8) | |
| 3 | 79 (5.2) | |
| 4 | 39 (3.5) | |
| 5 or more | 84 (4.2) | |
| Year of hospitalisation | | 0.002 |
| 2015 | 276 (7.3) | |
| 2016 | 216 (6.1) | |
| 2017 | 194 (5.2) | |
| 2018 | 217 (5.9) | |
| Number of drugs at hospital discharge | 10.91 (SD=3.89) | <0.001* |

*Mann-Whitney U test.
CHOP, Swiss Classification of Surgical Interventions; ICD-10, International Classification of Diseases, 10th version.

was 9.07 (SD=3.32), with means of 10.91 (SD=3.89) drugs for patients discharged to a nursing home vs 8.95 (SD=3.24) for those discharged home. Online supplemental tables 1 and 2 present descriptive statistics of the older adult inpatients' health statuses and drugs prescribed at discharge.

## Associations between unplanned nursing home admission, sociodemographic characteristics, and the prevalence of clinical and medical conditions

We found a prevalence of older adults discharged to unplanned nursing home admission of 6.1% (n=903/N=14 705) over the whole time period, with a slight decrease in prevalence going forward (7.3% (n=276) in 2015 to 5.9% (n=217) in 2018). Bivariate associations showed that men had a lower prevalence of unplanned nursing home admission than women (4.0% (n=328) vs 8.8% (n=575)), as did subjects aged 65–69 years old (2.2%; n=49) compared with those 70–79 years old, 80–89 years old, and especially the oldest group, aged 90 or more (3.2% (n=192), 8.3% (n=437) and 19.7% (n=225), respectively).

Being concomitantly affected by several diseases increased the prevalence of unplanned nursing home admission, from 1.8% (n=5) for older adults with a single disease (ICD-10) to 6.8% (n=797) for those with five or more diseases. Furthermore, the number of surgical interventions was negatively associated with the prevalence of unplanned nursing home admission. Patients who had not undergone surgery showed a higher probability of unplanned nursing home admission (7.8%; n=379) than those who had undergone several interventions (3.5% (n=39) for four interventions, 4.2% (n=84) for five interventions) (table 1). The number of drugs prescribed at hospital discharge showed a positive linear relationship with unplanned nursing home admission (gamma=0.368) (figure 1).

## Associations between unplanned nursing home admission and drugs

Bivariate associations showed that drugs were also related to unplanned nursing home admission (table 2). On average, home-dwelling older adults discharged to a

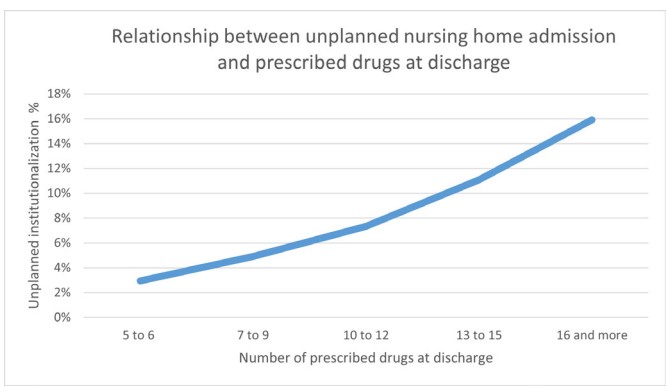

**Figure 1** Relationship between unplanned nursing home admission and number of prescribed drugs at discharge.

**Table 2** Prevalence of unplanned nursing home admission among polymedicated hospitalised older adults (N=14 705) with regard to associations with different classes of prescribed drugs

| Drugs (ATC code) | Unplanned nursing home admission | | |
| --- | --- | --- | --- |
| | No drugs in this class n (%) | Drugs in this class n (%) | P value |
| *First level, main anatomical group* | | | |
| Blood and blood-forming organ drugs (B) | 180 (5.4) | 723 (6.4) | 0.050 |
| Dermatologicals (D) | 828 (5.8) | 75 (14.1) | <0.001 |
| Genitourinary system and sex hormones (G) | 737 (6.1) | 166 (6.3) | 0.699 |
| Systemic hormonal preparations, excluding sex hormones and insulins (H) | 737 (6.1) | 166 (6.5) | 0.403 |
| Anti-infectives for systemic use (J) | 736 (6.4) | 167 (5.3) | 0.020 |
| Antineoplastic and immunomodulating agents (L) | 881 (6.3) | 22 (3.5) | 0.005 |
| Drugs for the musculoskeletal system (M) | 815 (6.4) | 88 (4.3) | <0.001 |
| Antiparasitic products, insecticides and repellents (P) | 893 (6.2) | 10 (4.0) | 0.144 |
| Respiratory system drugs (R) | 771 (6.3) | 132 (5.5) | 0.147 |
| Sensory organ drugs (S) | 752 (5.5) | 151 (13.4) | <0.001 |
| *Second level, therapeutic subgroup* | | | |
| Stomatological preparations (A01) | 899 (6.1) | 4 (7.5) | 0.669 |
| Drugs for acid-related disorders (A02) | 384 (5.8) | 519 (6.4) | 0.136 |
| Drugs for functional gastrointestinal disorders (A03) | 805 (5.9) | 98 (9.8) | <0.001 |
| Antiemetics and antinauseants (A04) | 884 (6.1) | 19 (18.6) | <0.001 |
| Bile and liver therapy drugs (A05) | 900 (6.1) | 3 (7.9) | 0.652 |
| Drugs for constipation (A06) | 605 (4.8) | 298 (13.5) | <0.001 |
| Antidiarrhoeals, intestinal anti-inflammatory/anti-infective agents (A07) | 863 (6.0) | 40 (9.4) | 0.005 |
| Digestives, including enzymes (A09) | 883 (6.1) | 20 (8.4) | 0.148 |
| Diabetes drugs (A10) | 804 (6.6) | 99 (3.9) | <0.001 |
| Vitamins (A11) | 801 (6.2) | 102 (5.9) | 0.629 |
| Mineral supplements (A12) | 513 (4.8) | 390 (9.6) | <0.001 |
| Other alimentary tract and metabolism products (A16) | 901 (6.1) | 2 (5.9) | 0.950 |
| Cardiac therapy drugs (C01) | 792 (6.1) | 111 (6.3) | 0.792 |
| Antihypertensives (C02) | 888 (6.2) | 15 (4.6) | 0.237 |
| Diuretics (C03) | 621 (5.5) | 282 (8.1) | <0.001 |
| Peripheral vasodilators (C04) | 901 (6.1) | 2 (4.2) | 0.568 |
| Vasoprotectives (C05) | 884 (6.1) | 19 (7.2) | 0.471 |
| Beta-blocking agents (C07) | 588 (7.2) | 315 (4.8) | <0.001 |
| Calcium channel blockers (C08) | 762 (6.1) | 141 (6.1) | 0.964 |
| Agents acting on the renin-angiotensin system (C09) | 472 (7.2) | 431 (5.3) | <0.001 |
| Lipid-modifying agents (C10) | 720 (8.2) | 183 (3.1) | <0.001 |
| Anaesthetics (N01) | 898 (6.1) | 5 (13.5) | 0.061 |
| Analgesics (N02) | 158 (3.6) | 745 (7.2) | <0.001 |
| Antiepileptics (N03) | 753 (5.7) | 150 (10.3) | <0.001 |
| Drugs against Parkinson's disease (N04) | 814 (5.7) | 89 (18.1) | <0.001 |
| Psycholeptics (N05) | 201 (2.4) | 702 (11.0) | <0.001 |
| Psychoanaleptics (N06) | 565 (4.8) | 338 (11.9) | <0.001 |
| Other nervous system drugs (N07) | 881 (6.1) | 22 (5.9) | 0.813 |

ATC, Anatomical Therapeutic Chemical.

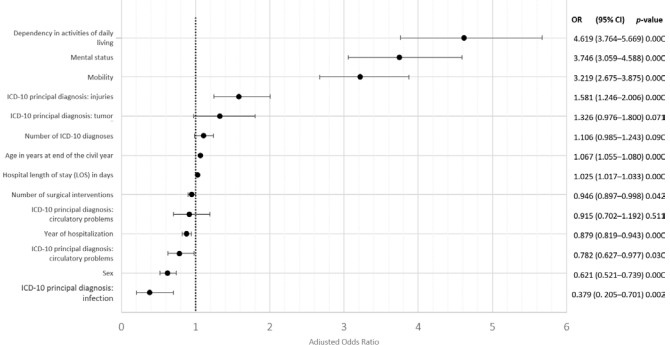

**Figure 2** Baseline GEE logistic regression model with unplanned nursing home admission as the dependent variable associated with sociodemographic, hospitalisation, and independent clinical and medical variables (N=14 705 observations for 9430 different subjects). GEE, generalised estimating equations; ICD-10, International Classification of Diseases, 10th version.

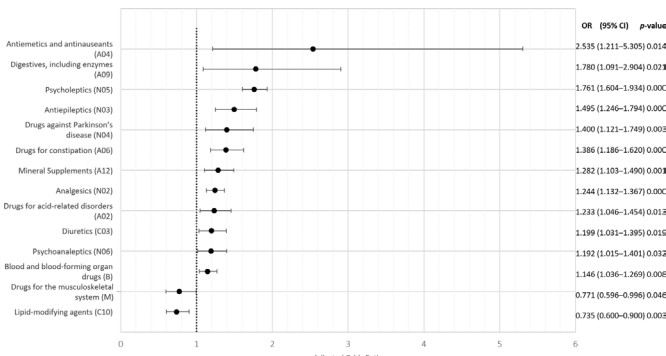

**Figure 3** The GEE logistic regression model of the drugs prescribed to older adults at discharge with significant predictive values (ORs) for unplanned nursing home admission (N=14 705 observations for 9430 different subjects)—controlled for the parameters of the baseline model. GEE, generalised estimating equations.

nursing home had more prescribed drugs than those returning to their home (10.9 (SD=3.9) drugs vs 8.9 (SD=3.2)). Psycholeptics (antipsychotics, anxiolytics, hypnotics and sedatives) and psychoanaleptic drugs (antidepressants, psychostimulants, nootropics and anti-dementia drugs), antiemetics and antinauseants, anti-Parkinson's disease drugs, and drugs treating constipation and the sensory organs were significantly associated with unplanned nursing home admission. On the contrary, patients taking lipid-modifying agents were less prone to unplanned nursing home admission.

### Multivariate baseline model

A baseline, GEE logistic regression model, including sociodemographic information, clinical data and diseases, was computed to predict unplanned nursing home admission among discharged polymedicated older adult patients (figure 2 and online supplemental table 3); prescribed drugs at hospital discharge were not included. If the 95% CI does not overlap the null value (eg, OR=1), then the higher the OR, the more the variable contributes to unplanned nursing home admission. Men had a lower probability of unplanned nursing home admission than women (OR=0.62; 95% CI 0.52 to 0.73). Patients' probability of unplanned nursing home admission increased with age (OR=1.07 for each additional year of age; 95% CI 1.05 to 1.08). Impaired mobility, dependency in the ADL and cognitive impairment revealed their substantial impacts on unplanned nursing home admission (OR=3.22; 95% CI 2.67 to 3.87; OR=4.62; 95% CI 3.76 to 5.67; and OR=3.75; 95% CI 3.06 to 4.59, respectively). Circulatory and infectious diseases were related to lower probabilities of unplanned nursing home admission (OR=0.78; 95% CI 0.63 to 0.98, and OR=0.38; 95% CI 0.20 to 0.70, respectively), whereas traumatic injuries were related to higher probabilities (OR=1.58; 95% CI 1.25 to 2.01). The number of ICD-10 diagnoses alone had no significant impact on the odds of unplanned nursing home admission (OR=1.11; 95% CI 0.98 to

1.24), in contrast to the number of surgical interventions undergone (CHOP), which was a protective factor against unplanned hospitalisation (OR=0.95; 95% CI 0.90 to 0.99). The year of hospital stay also had a significant impact, with more recent stays having lower probabilities of unplanned nursing home admission (OR=0.88; 95% CI 0.82 to 0.94, per ensuing year).

### Prediction of unplanned nursing home admission and drug prescription

A higher number of prescribed drugs were associated with a higher probability of unplanned nursing home admission (OR=1.17; 95% CI 1.15 to 1.19). Figure 3 and online supplemental table 4 present the baseline GEE logistic regression model shown in figure 2 completed with those drugs prescribed to older adults at discharge that had a significant statistical association (p<0.05) with unplanned nursing home admission. Drugs without a significant statistical association are not presented in figure 3 for simplification purposes. Antiemetics and antinauseants (OR=2.53; 95% CI 1.21 to 5.30 for each additional unit), digestives (OR=1.78; 95% CI 1.09 to 2.90), psycholeptics (OR=1.76; 95% CI 1.60 to 1.93), antiepileptics (OR=1.49; 95% CI 1.25 to 1.79) and anti-Parkinson's disease drugs (OR=1.40; 95% CI 1.12 to 1.75) were strongly linked to unplanned nursing home admission after controlling for other parameters. On the contrary, taking lipid-metabolism-modifying agents was associated with lower probabilities of unplanned nursing home admission (OR=0.73; 95% CI 0.60 to 0.90, for each extra drug from this class prescribed).

### Combined drug intake and probabilities of unplanned nursing home admission

To reduce collinearity and simplify the results, the combined intake of different ATC drug classes was recoded as a dichotomised variable for each drug pairing and added to the previous model.[22] Only the drugs and drug combinations prescribed to older adults at discharge that had significant associations (p<0.05)

with unplanned nursing home admission are presented. The combined intake of cardiac therapy and psychoanaleptic drugs was significantly associated with unplanned nursing home admission (OR=1.87; 95% CI 1.11 to 3.16), as were psychoanaleptics and diabetes drugs combined (OR=1.75; 95% CI 1.03 to 2.98), and psycholeptic drugs and vitamins combined (OR=1.71; 95% CI 1.03 to 2.84). On the contrary, the combined intake of beta-blocking agents and antiepileptics strongly diminished the odds of unplanned nursing home admission (OR=0.39; 95% CI 0.23 to 0.67).

We also investigated the risk of unplanned nursing home admission for combined drug intake within the same drug class. The combined intake of two or more antiemetic and antinauseants (OR=2.65; 95% CI 1.26 to 5.58), psycholeptics (OR=1.64; 95% CI 1.46 to 1.85), antiepileptics (OR=1.55; 95% CI 1.23 to 1.96) or anti-Parkinson's disease drugs (OR=1.44; 95% CI 1.13 to 1.83) was strongly associated with a higher probability of unplanned nursing home admission.

Online supplemental table 5 summarises the main findings from our predictive analysis.

## DISCUSSION

This population-based hospital registry study used longitudinal data to examine the unplanned nursing home admission of hospitalised polymedicated older inpatients, revealing a 6.1% prevalence rate over the 4-year dataset, in agreement with previous work by Luppa *et al* (men: 5.4%; women: 6.0%) and Goodwin *et al* (5.5%).[7 28] The slight decrease in prevalence over the 4 years of the study may be explained by improvements in the regional home care services' contribution to maintaining older adults at home, but also to planned nursing home admissions without the requirement for intermediate hospitalisation.[29] Furthermore, the number of places in the region's nursing homes increased in that period,[30] allowing people for whom care at home became impossible to be admitted to a nursing home more promptly.

Our predictive analysis revealed that the group of the oldest adults, presenting functional mobility impairments, dependency in the ADL and cognitive impairment, was also at a high risk of unplanned nursing home admission, which is consistent with previous retrospective and prospective studies.[31 32] Very old inpatients (≥90 years old) were much more likely to have an unplanned nursing home admission than those aged 65–69 years (19.7% vs 2.2%). This finding was expected and matched with previous research,[33] bearing in mind that the very oldest group presented with a high prevalence of multimorbidity and advanced functional and cognitive impairments. Unexpectedly, regardless of age, our results showed that older women had a higher prevalence and probability of unplanned nursing home admission than men.[34] Yet, our data could not entirely explain this result. Previous publications have indicated that social and life expectancy factors may play roles in the different rates of nursing home admission between older adult men and women.[34]

Our findings highlighted that functional and cognitive impairments were strong risk factors for unplanned nursing home admission, which is in line with the studies by Luppa *et al* and Goodwin *et al*.[28 34] Likewise, our results emphasised a high risk of unplanned nursing home admission among non-surgically treated and trauma patients. This could be explained by the relationship between orthopaedic guidelines on traumatic injuries among older adults that suggest avoiding surgery, for several medical reasons (number and severity of multimorbidities), and which may lead to increased functional impairment and unplanned nursing home admission, as suggested by Gardner *et al* and Cutugno.[35 36]

As might be expected, older adults who underwent an unplanned nursing home admission had more prescribed drugs than those returning home. Our results were in line with the retrospective study by Lucchetti *et al*, which demonstrated a relationship between the prescription of cardiovascular, gastrointestinal, and metabolic drugs and unplanned nursing home admission.[37]

Our findings indicated that patients prescribed more than one drug from the same class of drugs—from the classes of antiemetics and antinauseants, psycholeptics, antiepileptics or anti-Parkinson's disease drugs—had a higher risk of discharge to a nursing home. Although this phenomenon is still underinvestigated, our findings are not in line with the few existing studies in this area, which have presented no significant relationships between drug interactions and unplanned nursing home admission.[38] However, in hospital settings, a recent systematic review reported drug–drug interactions among 80% or more of older inpatients.[39] Since polymedicated older inpatients should be considered as a population at a high risk of adverse outcomes, further studies should investigate how drug–drug interactions might predict the risks of nursing home admission.

Our findings undeniably mirrored existing evidence that chronic conditions and debilitating comorbidities are significant risk factors for unplanned nursing home admission.[3 7] However, they also raised questions regarding hospitalisation's effects on the individual ageing process, which likely interact to produce a cascade of factors towards functional decline and dependency.[1] The adverse effects of hospitalisation begin immediately and progress rapidly.[1] Harrison *et al* and Haaksma *et al* described ways in which acute and exacerbated acute and chronic disorders, reinforced by existing undiagnosed geriatric syndromes (frailty, delirium, pressure sores, functional incontinence), contributed to hospitalised older patients being unable to return home and needing to be discharged to a nursing home.[2 16] Previous studies suggested that silent geriatric syndromes such as frailty and functional decline, together with polypharmacy, are not only clinically characteristic of older adults but also potential predictors of being at risk of a further loss of independence and subsequent nursing home admission.

Montes *et al* pointed out the dramatic rise in numbers of frail, hospitalised older adults. This increase generates concerns about whether nursing homes—already suffering from long admission waiting lists of home-dwelling older adults—will be able to cope with older adults' complex care needs.[40]

Although some of the predisposing predictors identified cannot be treated (ie, sex, age), they may still contribute to an older adult's risk of being discharged to a nursing home and subsequently exacerbate their situation there. Given that hospitalisation introduces stressors that may increase the chances of unplanned nursing home admission,[41 42] using patients' electronic hospital data could help identify the high-risk older adults who would benefit from specific preventive interventions. Being able to rapidly identify inpatients at a high risk of unplanned nursing home admission may help professional caregivers to provide them with the appropriate community health resources, such as community-based rehabilitation programmes. This would help older people to remain in their community for longer.

### Study strengths and limitations
Although our population-based study's findings could be generalised to other regions of Switzerland, any interpretations should be made with caution. The Swiss Federal Statistical Office collects minimal annual data from public and private hospitals (number of hospitalisations, ICD diagnoses, length of stay, place of discharge, age and sex), but these indicated that our data were similar to those from other cantons with analogous healthcare structures.[43] However, we did not have access to more detailed data with which to compare with our dataset and explore potential biases or significant differences. Nevertheless, the Valais Hospital is the third largest hospital in Switzerland with more than 1000 beds and over 35 000 hospitalisations per year. Therefore, our findings could provide information to help better define which integrated healthcare approaches could be implemented to attenuate the risk factors associated with unplanned nursing home admission following an acute hospital admission or readmission. The numerous predictors revealed in our study enabled us to conceptualise an overview of hospitalised older adults' health conditions before their unplanned nursing home admission. As healthcare moves towards ever-more personalised medicine, this result could help create more refined, tailored, future interventions via 'risk profiles' defined using each older adult's personal predictors.

Our study had some limitations. The absence of data on patients' functional status before hospital admission meant that we could not assess changes to that status during hospitalisation, such as the influence of the development or deterioration of functional and cognitive impairment. We did not compute analysis on specific disorders such as neurodegenerative diseases like dementia and Parkinson's disease because this was beyond the scope of our study protocol. However, further

analyses could confirm earlier studies showing that these diseases significantly affect a person's risk of nursing home admission after hospitalisation, with almost 90% of patients with dementia being admitted into a nursing home before dying.[15 16] Additionally, our dataset was based on routinely collected data, and we were unable to control for potential data assessment errors made by the Valais Hospital's healthcare staff at discharge. Moreover, we were unable to assess deceased patients' death certificates as these were unavailable and beyond the scope of our study. Although the study considered statistical associations between drugs and unplanned nursing home admission, it did not use clinically diagnosed drug–drug interactions. Lastly, our data were unable to identify hospitalisations that might have been triggered by limited care options at home or hospitalisations that were necessary while awaiting a place in a nursing home. These cases of planned nursing home admissions could not be distinguished from the unplanned nursing home admissions considered in the study. In addition, some patients may not have been transferred directly from hospital to nursing homes and may have had to stay in an intermediate structure while awaiting a place. These patients were not included in the study due to the unavailability of this information in the database.

### CONCLUSION
The sociodemographic characteristics of hospitalised older inpatients, together with their clinical and medical conditions and their prescribed drugs, can provide us with a significant set of risk factors for unplanned nursing home admission, sustaining our stated hypotheses. Identifying these risk factors for unplanned nursing home admission could be of great assistance in developing predictive tools and tailored intervention programmes aimed at reducing the number of older adults placed in nursing homes. Our results showed that the patient-related risk factors leading to nursing home admission were based on declines in physical and cognitive function. Treatment with single drugs and combinations of drugs was also associated with unplanned nursing home admission, indicating that multiple chronic health conditions are important risk factors of a non-return home. Our findings may help to identify those older inpatients at the greatest risk of unplanned nursing home admission, enabling their care to be optimised by counterbalancing those risk factors. Further research is required across large samples of older inpatients to investigate whether tailored interventions at early stages in chronic diseases could delay physical and cognitive dysfunction and reduce unplanned nursing home admissions among this growing segment of the population.

**Author affiliations**
[1]Institute of Biomedical Sciences Abel Salazar, University of Porto, Porto, Portugal
[2]School of Health Sciences, HES-SO Valais Wallis, Sion, Switzerland
[3]Département de Psychiatrie, Centre Hospitalier Universitaire Vaudois, Prilly, Switzerland

[4] Institute of Psychology, Research Center for the Psychology of Health, Aging and Sports Examination, University of Lausanne, Lausanne, Switzerland
[5] Institute for Primary Health Care (BIHAM), University of Bern, Bern, Switzerland
[6] Clinical Pharmacology and Toxicology, Clinical of General Internal Medicine, Inselspital - University Hospital of Bern, Bern, Switzerland
[7] Escola Superior de Enfermagem do Porto, Porto, Portugal
[8] FORS, Swiss Centre of Expertise in the Social Sciences, University of Lausanne, Lausanne, Switzerland

**Acknowledgements** The authors thank the partner hospital, including the hospital's data warehouse, for its valuable contributions.

**Contributors** BW, FP and HV had the original idea. BW, MdRC, MMM and HV provided conceptual and methodological expertise to the study design, and BW, FP, CM-M, AvG and HV contributed to data analysis and interpretation. BW, FP and HV were major contributors to writing the manuscript. HV is the guarantor of the study. All authors read, edited and approved the final manuscript.

**Funding** This work was supported by the Swiss National Science Foundation (grant number 407440_183434/1).

**Competing interests** None declared.

**Patient consent for publication** Not required.

**Ethics approval** This study involves human participants and ethical approval was obtained from the Human Research Ethics Committee of the Canton of Vaud (CER-VD, 2018-02196), and this permitted the partnering hospital's data warehouse to provide the appropriate dataset. Given the retrospective data source, obtaining consent from the patients concerned was impossible or posed disproportionate difficulties. The present study respects the legal requirements for research projects involving data reuse without consent, as set out in Article 34 from the Swiss Human Research Act (HTA).

**Provenance and peer review** Not commissioned; externally peer reviewed.

**Data availability statement** As part of the Data Use Agreement, authors are not allowed to provide raw data. Upon a reasonable request, the corresponding author will provide statistical programming code used to generate results

**ORCID iDs**
Filipa Pereira http://orcid.org/0000-0001-9207-4856
Henk Verloo http://orcid.org/0000-0002-5375-3255

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
