## [Reviewer comments · BMJ Open]

ARTICLE DETAILS

TITLE (PROVISIONAL)	Unplanned nursing home admission among discharged polymedicated older inpatients: a single-centre, registry-based study in Switzerland
AUTHORS	Pereira, Filipa; Verloo, Henk; von Gunten, Armin; del Río Carral, María; Meyer-Masseti, Carla; Martins, Maria Manuela; Wernli, Boris

VERSION 1 – REVIEW

REVIEWER	Fabbietti, Paolo INRCA
REVIEW RETURNED	26-Oct-2021

GENERAL COMMENTS	Dear authors, - in table 1 is better to delete % from the cells. Is sufficient to show it in the name of the column. - at the end of table 1 is better to show each year in a single separated line - you can study other variables like sarcopenia for example or other that you can see in this paper (that you can please cite): "Association between hospitalization-related outcomes, dynapenia and body mass index: The Glisten Study". Rossi et al
--

REVIEWER	Marashi-Pour, Sadaf NSW Health, Bureau of Health Information
REVIEW RETURNED	29-Oct-2021

GENERAL COMMENTS	Thank you for providing your paper for review. This work has looked at factors associated with unplanned institutionalization following acute hospitalisations among older patients and could be of great value aiming to improve care among older patients. The study has some limitations that need to be clearly addressed and could benefit from few further analyses. Please see below my detailed comments: • Not having the information regarding hospitalizations, transfers and rehospitalizations to other hospitals, except for the Valais hospital could be a big limitation for this study. It is possible that a prior hospitalisation in another hospital (e.g. a prior surgery) have had an effect on the unplanned institutionalization or a patient might be discharged to nursing home following a transfer to another hospital for a short service. It is not clear how these issues have been taken into account. For example, to reduce its effect, patients coded as transferred in/transferred out (if this information was available at admission/discharge) could be excluded from the analyses, after checking any potential bias that their exclusions
---

	might cause If linked data including the entire patients' pathway is not used it needs to be highlighted as a limitation of the study.  • The unplanned institutionalizations need to be defined more clearly based on the available data elements. For example, was a code available in the data regarding the discharge position of the patients following their acute admissions at Valais hospital which was used? • At page 7 row 200, It is not clear why the modelling analysis included three of the health status variables (e.g. based on univariate analyses?). • It is not indicated how the multivariable models were developed (e.g. any stepwise modelling approaches used?) • It is mentioned that GEE regression model was used but some details are not provided, including why the data was correlated (e.g. repeated hospitalisations for the same patients?), or what is considered in the model as the within-group correlation structure. • It is not clear if there were any deaths and if yes how deaths were taken into account. If this information wasn't available and /or wasn't taken into account, it needs to be highlighted as one of the limitations of the study. Some of the observed associations might be explained by deaths, if not taken into account using methods such as competing risks regression models. • In the results section at page 9 row 264, it is not clear what "g" is referring to. • The results mostly only include percentages, and lacking numbers. I think it's better that numbers be added to the results. • In the results section end of page 9, the odds ratios aren't described properly. • Table 3 at page 21 needs reformatting as currently is hard to read • Page 10 at results section lines 299-301 is not very clearly written and perhaps some rewording could be helpful. • Some of the figures/tables did not have a title so was not clear which of these tables/figures are being referenced in the text
--	--

REVIEWER	Simões, Pedro Augusto Universidade da Beira Interior Faculdade de Ciencias da Saude
REVIEW RETURNED	07-Nov-2021

GENERAL COMMENTS	The Introduction should be shorter and more concise, some paragraphs should be in the Discussion section and not in the Introduction. The Methods should also be made more clear. The description of the population sample should be in the beginning of the Results section and not in the Methods (make it harder to read). In the Results you repeat almost all the information that are in the tables and figures in the text (they should be complementary and not repetition of the same information). In table 1 you should mention the number of people in each group (eg. Female and Male...) and not only the percentage so we can better understand the size difference between them. You also should mention the values obtained even the ones that were not statistically significant (the same for table 2). As mentioned before, there shouldn't be repeat information. Therefore, table 3 or figures 1 and 2 should be removed since it is the same information in both places. You can choose to present
--

	the information between tables or figures, but do not present the same information in both.
--	---

VERSION 1 – AUTHOR RESPONSE

Reviewer: 1
Dr. Paolo Fabbietti, INRCA

Reviewers' 1 comments	Response by authors	Location in text
- in table 1 is better to delete % from the cells. Is sufficient to show it in the name of the column.	We have adapted Table 1 as suggested.	Table 1
- at the end of table 1 is better to show each year in a single separated line	We have adapted Table 1 as suggested.	Table 1
- you can study other variables like sarcopenia for example or other that you can see in this paper (that you can please cite): "Association between hospitalization-related outcomes, dynapenia and body mass index: The Glisten Study". Rossi et al	Indeed, it would have been clinically relevant to explore specific clinical syndromes or diseases like sarcopenia or dynapenia. Unfortunately, this was impossible for two main reasons: 1) Our study fell within the framework of a nationally funded research programme in Switzerland (NRP 74). Specific clinical syndromes or medical diagnostics did not fall within the research programme's focus, nor that of this publication, but they could be the subjects of further publications. 2) We used retrospective data from a longitudinal, population-based hospital registry of electronic health records. Despite this limitation, the use of routinely collected data is considered an added value to predictive analysis and should not be considered as a gap. Some examples are: - Andrew Davy, Thomas Hill, Sarahjane Jones, Alisen Dube, Simon c Lea, Keiar I Watts, M d Asaduzzaman, A predictive model for identifying patients at risk of delayed transfer of care: a retrospective, cross-sectional study of routinely collected data, International Journal for Quality in Health Care, Volume 33, Issue 3, 2021, https://doi.org/10.1093/intqhc/mzab130 - Daniels H, Hollinghurst J, Fry R, Clegg A, Hillcoat-Nallétamby S, Nikolova S, Rodgers SE, Williams N, Akbari A. The Value of Routinely Collected Data in Evaluating Home Assessment and Modification Interventions to Prevent Falls in Older People: Systematic Literature Review. JMIR Aging.	

	2021 Apr 23;4(2):e24728. https://aging.jmir.org/2021/2/e24728	
--	---	--

Reviewer: 2

Ms. Sadaf Marashi-Pour, NSW Health

Reviewers' 2 comments	Response by authors	Location in text
1. Thank you for providing your paper for review. This work has looked at factors associated with unplanned institutionalization following acute hospitalisations among older patients and could be of great value aiming to improve care among older patients. The study has some limitations that need to be clearly addressed and could benefit from few further analyses. Please see below my detailed comments:	We thank the reviewer for her support.	
2. Not having the information regarding hospitalizations, transfers and rehospitalizations to other hospitals, except for the Valais hospital could be a big limitation for this study. It is possible that a prior hospitalisation in another hospital (e.g. a prior surgery) have had an effect on the unplanned institutionalization or a patient might be discharged to nursing home following a transfer to another hospital for a short service. It is not clear how these issues have been taken into account. For example, to reduce its effect, patients	Thank you for your interesting comment. Fortunately, the hospital registry contained information on where patients had arrived from and where they were discharged to, allowing us to correctly identify our population of interest and customise our dataset. To better explain this, we have adapted the manuscript as follows: “Since where patients had arrived from and where they were discharged to were two distinct variables, the dataset was recoded and customised to identify the number of older inpatients admitted straight from their home and then discharged to a nursing home, as presented in a previous paper [29].”	Dataset customizing for predictive analysis Lines 187-190

coded as transferred in/transferred out (if this information was available at admission/discharge) could be excluded from the analyses, after checking any potential bias that their exclusions might cause. If linked data including the entire patients' pathway is not used it needs to be highlighted as a limitation of the study.		
3. The unplanned institutionalizations need to be defined more clearly based on the available data elements. For example, was a code available in the data regarding the discharge position of the patients following their acute admissions at Valais hospital which was used?	Indeed, this is a very important element to specify. We have completed the manuscript to clarify how we obtained information on unplanned nursing home admissions: “Since where patients had arrived from and where they were discharged to were two distinct variables, the dataset was recoded and customised to identify the number of older inpatients admitted straight from their home and then discharged to a nursing home, as presented in a previous paper [29]. Each subject’s unique identifier was used to distinguish between different observations from 2015 to 2018 and to account for hospital readmissions. (...) Unplanned nursing home admission after discharge from our participating hospital between 2015 and 2018 was identified by the difference between the original abode (home) and the destination at discharge, and this was recoded as the dependent variable of interest.”	Population and data collection Lines 187-197
4. At page 7 row 200, It is not clear why the modelling analysis included three of the health status variables (e.g. based on univariate analyses?).	These three variables (mobility, dependency in the ADL, and cognitive status) were retained in the modelling analysis because a descriptive analysis revealed them to be significant. To make this more explicit, we have adapted the text as follows: “Numerous variables were used to describe older adults’ health status at	Dataset customizing for predictive analysis Health Variables Lines 204-208

	the end of their hospital stay. The modelling analysis included three of the six hierarchical clusters preliminarily computed as being variables significantly associated with more unplanned nursing home admissions in the descriptive analysis: mobility, dependency in the ADL and cognitive status [29].”	
5. It is not indicated how the multivariable models were developed (e.g. any stepwise modelling approaches used?)	In response to this comment, we have added the following sentence in the Data analysis strategy section: “A multiple bivariate logistic regression analysis was conducted using cross-tabulations to investigate whether the sociodemographic, health and drugs data (more than one independent variable) significantly predicted unplanned nursing home admission from 2015 to 2018 (our single dichotomous outcome).	Data analysis strategy Lines 236-239
6. It is mentioned that GEE regression model was used but some details are not provided, including why the data was correlated (e.g. repeated hospitalisations for the same patients?), or what is considered in the model as the within-group correlation structure.	To answer this request, we have completed the manuscript as follows: “In a second stage, a series of generalised estimating equations (GEE or population-averaged logistic regression models) were computed to predict how sets of predictors influenced the probability of unplanned nursing home admission. The variables included were derived from the significant associations between sociodemographic characteristics, clinical and medical conditions and unplanned nursing home admission (Table 1). This baseline model was completed using the drugs prescribed to older inpatients who underwent an unplanned nursing home admission. Lastly, based on our literature review, known drug–drug interactions between different ATC drug classes were added to the baseline model. The model estimated each predictor’s impact, other things being equal, by estimating its net impact controlling for	Data analysis strategy Lines 242-251

	confounding factors (adjusted odds ratios).”	
7. It is not clear if there were any deaths and if yes how deaths were taken into account. If this information wasn't available and /or wasn't taken into account, it needs to be highlighted as one of the limitations of the study. Some of the observed associations might be explained by deaths, if not taken into account using methods such as competing risks regression models.	As the manuscript explains, “ all the patients included in the study followed a home to hospital to long-term residential care facility pathway ” (lines 169-171). “ Since where patients had arrived from and where they were discharged to were two distinct variables, the dataset was recoded and customised to identify the number of older inpatients admitted straight from their home and then discharged to a nursing home ” (lines 187-189). Therefore, older adults who died during hospitalisation were automatically excluded, as were those who returned to their homes.	Population and data collection (lines 169-171) Dataset customizing for predictive analysis (lines 187-189)
8. In the results section at page 9 row 264, it is not clear what “g” is referring to.	Thank you for alerting us to this relevant point. The “g” refers to the Gamma Test, or Kruskal and Goodman's Gamma Test, a non-parametric correlation test that is a non-parametric alternative to Pearson's correlation test. This test was chosen because of the ordinal nature of the dependent variable (number of drugs prescribed at hospital discharge). To make this point clearer for readers we have written “gamma” instead of “g”: “The number of drugs prescribed at hospital discharge showed a positive linear relationship with unplanned nursing home admission (gamma = 0.368).”	Results Lines 295-297
9. The results mostly only include percentages, and lacking numbers. I think it's better that numbers be added to the results.	We have updated our results by adding absolute numbers to all the percentage prevalence values reported in the manuscript.	Whole manuscript
10. In the results section end of page 9, the odds ratios aren't described properly.	We agree with the reviewer that the data (averages and not odds ratios) were not presented adequately. To avoid ambiguity, we have completed this section as follows: “On average, older adults whose discharge to a nursing home was unplanned had more prescribed drugs	Results Lines 301-304

	than those returning home [10.9 (SD = 3.9) drugs vs 8.9 (SD = 3.2)].”	
11. Table 3 at page 21 needs reformatting as currently is hard to read	Thank you for your comment. Table 3 presents a summary of the predictive analysis, and this is why the presentation is less standardised. To facilitate readability, we have distinguished between the risk factors and protective factors of unplanned nursing home admission. Furthermore, the different factors are classified in decreasing magnitude of odds ratio.	Table 3 (now a supplementary table)
12. Page 10 at results section lines 299-301 is not very clearly written and perhaps some rewording could be helpful.	We believe that the answer to question 6 (in the data analysis section) brings more clarity to this sentence in the results section. Nevertheless, we have tried to reformulate it as follows: “Figure 3 and Supplementary Table 3 present the baseline GEE logistic regression model shown in Figure 2 completed with those drugs prescribed to older adults at discharge that had a significant statistical association ($p < 0.05$) with unplanned nursing home admission.”	Prediction of unplanned nursing home admission and drug prescription Lines 336-339
13. Some of the figures/tables did not have a title so was not clear which of these tables/figures are being referenced in the text	A list of the figure legends appears just before the reference section, as can be seen on page 17 (line 507), and above each of the two tables (pages 20–22).	

Reviewer: 3

Dr. Pedro Augusto Simões, Universidade da Beira Interior Faculdade de Ciencias da Saude

Reviewers' 3 comments	Response by authors	Location in text
The Introduction should be shorter and more concise, some paragraphs should be in the Discussion section and not in the Introduction.	We have revised and shortened the whole introduction as suggested by the reviewer.	Introduction
The Methods should also be made more clear. The description of the population sample should be in the beginning of the Results	We have made the amendment as suggested. A new section—Population description—now appears at the beginning of the Results.	Dataset customizing for predictive analysis Page 7-8

section and not in the Methods (make it harder to read).		Results Page 9-10
In the Results you repeat almost all the information that are in the tables and figures in the text (they should be complementary and not repetition of the same information).	We have simplified the text to avoid unnecessary repetition with the figures and tables. Nevertheless, since the journal is also intended for clinicians, we have intentionally left some text in addition to Figures 2 and 3, to support the interpretation and implementation of the GEE logistic regression model's findings.	Results
In table 1 you should mention the number of people in each group (eg. Female and Male...) and not only the percentage so we can better understand the size difference between them. You also should mention the values obtained even the ones that were not statistically significant (the same for table 2).	We agree with your suggestion, and we have completed Tables 1 and 2 with the absolute numbers beside the percentages.	Table 1 Table 2
As mentioned before, there shouldn't be repeat information. Therefore, table 3 or figures 1 and 2 should be removed since it is the same information in both places. You can choose to present the information between tables or figures, but do not present the same information in both.	To avoid this redundancy, Table 3—which presents a summary of the predictive analysis—is now a Supplementary File.	Table 3 moved to Supplementary Table 4

VERSION 2 – REVIEW

REVIEWER	Marashi-Pour, Sadaf NSW Health, Bureau of Health Information
REVIEW RETURNED	21-Jan-2022

GENERAL COMMENTS	Thank you for providing your revised paper for review. Please see below my further comments:  • I think currently the paper is too long and making it a little shorter could make it easier to follow. For example, maybe more emphasis on the adjusted associations in the text rather than the univariate results could help. • I think the definition of the unplanned nursing home admission and patients included in the analyses need more clarification. For example: How planned admissions to nursing home among the included patients could be distinguished? It is mentioned in the responses to my previous comments that patients returned home/deaths were excluded from the entire analyses? and all
--

	patients followed a home to hospital to long-term residential care facility. I think it is worth making this clearer in the paper including what a long-term residential care facility includes (e.g., one is nursing home which is the outcome of interest). And what was the reason for any of the exclusions, the number excluded, and any potential biases caused by the exclusions needs to be mentioned and discussed.  • Following the above comment, clarifications needed regarding the following sentence at line 252 of the results section: “On average, older adults whose discharge to a nursing home was unplanned had more prescribed drugs than those “returning home” [10.9 (SD = 3.9) drugs vs 8.9 (SD = 3.2)]”. • Not having access to the linked data and data from other hospitals and any potential biases caused by these and by any of the exclusions need to be highlighted more clearly in the discussion section. • I think the data analysis strategy section need some rewording to become clearer and shorter. For example:  - Please replace the word bivariate in the paper with the word univariate to more clearly refer to the unadjusted analysis conducted. For example, at line 199 something like “univariate analysis using logistic regression models was conducted to investigate...” instead of “multiple bivariate logistic regression analysis was conducted using cross-tabulations to investigate...”. Including the odds ratios in these tables could also be useful. And it seems that variables significantly associated with unplanned nursing home admission in the univariate analysis, were used to develop the multivariable models using GEE models. - Line 207: this sentence is not very clear: “This baseline model was completed using the drugs prescribed to older inpatients who underwent unplanned nursing home admission.” Does it mean that the base line model was completed by adding to it drugs that were found significantly associated with unplanned nursing home admission based on the univariate analysis? Did adding drugs to the model make any changes in the associations observed and reported in the baseline model in the previous figure/supplementary table? - How authors have ensured that their final multivariable model is not overfitted? How many parameters were included in the final multivariable model? - As is also indicated in the paper the GEE method is used to estimate population-averaged estimates. Please delete lines 213-216 (or please modify). • Line 242: with five or more diseases • Line 246, for investigating the association between number of drugs prescribed and unplanned nursing home admission in the univariate analysis authors could use logistic regression consistent with other univariate analyses performed. • In the discussion section line 324, please modify the wording to make it clear that the comparison being mentioned is related to the percentages in the unadjusted analysis, as with the current wording (i.e., tenfold higher risk) it could be confused with relative risks/odds ratios and the adjusted analyses. • Study strengths and limitations, line 378: authors have indicated that their findings regarding the single hospital, included in their study, could be generalised to other regions of Switzerland. Please include in the paper how authors came to this conclusion. Any representativeness analysis been conducted?
--	--

REVIEWER	Simões, Pedro Augusto
----------	-----------------------

	Universidade da Beira Interior Faculdade de Ciencias da Saude
REVIEW RETURNED	23-Dec-2021

GENERAL COMMENTS	In the tables 1 and 2, you should mention the values obtained even the ones that were not statistically significant. The figures mentioned in the manuscript were not present in the reviewed data available. References 13 (https://pubmed.ncbi.nlm.nih.gov/32065410/) and 40 (https://www.researchgate.net/publication/348919525_Predominant_factors_of_institutionalization_in_the_elderly_a_comparative_study_between_home_nursing_and_community_dwelling) should be corrected.
--

VERSION 2 – AUTHOR RESPONSE

Reviewer: 2

Ms. Sadaf Marashi-Pour, NSW Health

Reviewers' 2 comments	Response by authors	Location in text
Thank you for providing your revised paper for review. Please see below my further comments:  I think currently the paper is too long and making it a little shorter could make it easier to follow. For example, maybe more emphasis on the adjusted associations in the text rather than the univariate results could help. 	As requested by the reviewer, we shortened the univariate data in the results' section. The deleted section has been replaced by a supplementary table (Supplementary Table 1. Descriptive statistics of the older inpatients' health status).	Results Lines 227–234
 I think the definition of the unplanned nursing home admission and patients included in the analyses need more clarification. For example: How planned admissions to nursing home among the included patients could be distinguished? It is mentioned in the responses to my previous comments that patients returned home/deaths were excluded from the entire 	Thank you for your comment. This definition is key to understanding the manuscript. We hope it is much clearer now: “Our study defined ‘unplanned nursing home admission’ as the impossibility for a formerly home-dwelling older adult inpatient to return there after hospital discharge, and this included any new admission to a nursing home following an acute care admission [2]. All the patients included in the study followed home to hospital to nursing home pathway.”	Population and data collection Lines 140–144

analyses? and all patients followed a home to hospital to long-term residential care facility. I think it is worth making this clearer in the paper including what a long-term residential care facility includes (e.g., one is nursing home which is the outcome of interest). And what was the reason for any of the exclusions, the number excluded, and any potential biases caused by the exclusions needs to be mentioned and discussed.	To simplify, “long-term residential care facility” has been removed and replaced by “nursing home”. We completed the manuscript with the number and reasons of exclusions: “Since where patients had arrived from and where they were discharged to were two distinct variables, the dataset was recoded and customised to identify the number of older adult inpatients admitted straight from their homes and then discharged to a nursing home (n = 903) or returning to their homes (n = 13,802), as presented in a previous paper [24]. Therefore, older adults who died during hospitalisation (as assessed by the Valais Hospital’s healthcare staff) were automatically excluded (n = 131).” In addition, we completed the study limitations’ section as follows: “Additionally, our dataset was based on routinely collected data, and we were unable to control for potential data assessment errors made by the Valais Hospital’s healthcare staff at discharge. Moreover, we were unable to assess deceased patients’ death certificates as these were unavailable and beyond the scope of our study.”	Dataset customising for predictive analysis Synthesising the extracted data Lines 161–166 Study strengths and limitations Lines 409–412
 Following the above comment, clarifications needed regarding the following sentence at line 252 of the results section: “On average, older adults whose discharge to a nursing home was unplanned had more prescribed drugs than those “returning home” [10.9 (SD = 3.9) drugs vs 8.9 (SD = 3.2)]”. 	We have adapted this sentence as follows: “On average, home-dwelling older adults discharged to a nursing home had more prescribed drugs than those returning to their home [10.9 (SD = 3.9) drugs vs 8.9 (SD = 3.2)].”. This is mentioned in the Discussion section: “As might be expected, older adults who underwent an unplanned nursing home admission had more prescribed drugs than those returning home. Our results were in line with the	Results Lines 261–263

	retrospective study by Lucchetti et al., which demonstrated a relationship between the prescription of cardiovascular, gastrointestinal, and metabolic drugs and unplanned nursing home admission [37].”	Discussion Lines 349–352
 Not having access to the linked data and data from other hospitals and any potential biases caused by these and by any of the exclusions need to be highlighted more clearly in the discussion section. 	To address this relevant comment, the following statement was added in the Study strengths and limitations section: “The Swiss Federal Statistical Office collects minimal annual data from public and private hospitals (number of hospitalisations, ICD-diagnoses, length of stay, place of discharge, age and sex), but these indicated that our data were similar to those from other cantons with analogous healthcare structures [43]. However, we did not have access to more detailed data with which to compare with our dataset and explore potential biases or significant differences. Nevertheless, the Valais Hospital is the third largest hospital in Switzerland with more than 1,000 beds and over 35,000 hospitalisations per year. Therefore, our findings could provide information to help better define which integrated healthcare approaches could be implemented to attenuate the risk factors associated with unplanned nursing home admission following an acute hospital admission or readmission.”.	Study strengths and limitations Lines 388–397
 I think the data analysis strategy section need some rewording to become clearer and shorter. For example:  Please replace the word bivariate in the paper with the word univariate to more clearly refer to the unadjusted analysis conducted. For example, at line 199 something like “univariate analysis using logistic regression models was conducted to investigate...” instead of “multiple bivariate logistic regression analysis was conducted using cross-tabulations to investigate...”. 	We have adapted the manuscript to specify that bivariate analyses were unadjusted and multivariate analyses were adjusted. We have also clarified the description of the analytical part to be clearer. After discussing this as a team, we believe that including odds ratios would be redundant alongside percentages resulting from cross-tabulations, given the fact that those first	Data analysis strategy

Including the odds ratios in these tables could also be useful. And it seems that variables significantly associated with unplanned nursing home admission in the univariate analysis, were used to develop the multivariable models using GEE models.	results are unadjusted. The paper does present odds ratios for the multivariate-adjusted analysis.	
- Line 207: this sentence is not very clear: "This baseline model was completed using the drugs prescribed to older inpatients who underwent unplanned nursing home admission." Does it mean that the base line model was completed by adding to it drugs that were found significantly associated with unplanned nursing home admission based on the univariate analysis? Did adding drugs to the model make any changes in the associations observed and reported in the baseline model in the previous figure/supplementary table?	Thank you for your comment. We have revised our text as follows: "This adjusted baseline model was then completed by adding drugs that were found to be significantly associated with unplanned nursing home admissions in the previous analysis.". Adding these drugs did not cause any changes in Table 1.	Data analysis strategy Lines 212–214
- How authors have ensured that their final multivariable model is not overfitted? How many parameters were included in the final multivariable model?	We completed the Data analysis strategy section as follows: "The multivariable analysis model included 52 Level 2 ATC drug classes, respecting the good practices for logistical regressions involving large population-based samples [27]."	Data analysis strategy Lines 210–212
- As is also indicated in the paper the GEE method is used to estimate population-averaged estimates. Please delete lines 213-216 (or please modify).	As requested by the reviewer, lines 213 to 216 have been deleted.	Data analysis strategy Lines 217–220
• Line 242: with five or more diseases	We corrected as suggested: "Being concomitantly affected by several diseases increased the prevalence of unplanned nursing home admission, from 1.8% (n = 5) for	Results Lines 248–250

	older adults with a single disease (ICD-10) to 6.8% (n = 797) for those with five or more diseases.”	
 Line 246, for investigating the association between number of drugs prescribed and unplanned nursing home admission in the univariate analysis authors could use logistic regression consistent with other univariate analyses performed. 	As mentioned before, we believe that including odds ratios would be redundant alongside percentages resulting from cross-tabulations, given the fact that those first results are unadjusted. However, considering the reviewer’s comment, we have added the number of prescribed drugs in Table 1.	
 In the discussion section line 324, please modify the wording to make it clear that the comparison being mentioned is related to the percentages in the unadjusted analysis, as with the current wording (i.e., tenfold higher risk) it could be confused with relative risks/odds ratios and the adjusted analyses. 	Thank you for this important remark. We changed our wording as follows: “Very old inpatients (≥ 90 years old) had an almost tenfold higher risk 17.5% (19.7% vs 2.2%) more chance of an unplanned nursing home admission than those aged 65–69.”	Discussion Lines 333–334
 Study strengths and limitations, line 378: authors have indicated that their findings regarding the single hospital, included in their study, could be generalised to other regions of Switzerland. Please include in the paper how authors came to this conclusion. Any representativeness analysis been conducted? 	We completed the Study strengths and limitations section with the following statement: “The Swiss Federal Statistical Office collects minimal annual data from public and private hospitals (number of hospitalisations, ICD-diagnoses, length of stay, place of discharge, age and sex), but these indicated that our data were similar to those from other cantons with analogous healthcare structures [43]. However, we did not have access to more detailed data with which to compare with our dataset and explore potential biases or significant differences. Nevertheless, the Valais Hospital is the third largest hospital in Switzerland with more than 1,000 beds and over 35,000 hospitalisations per year.”	Study strengths and limitations Lines 388–394

Reviewer: 3

Reviewers' 3 comments	Response by authors	Location in text
*In the tables 1 and 2, you should mention the values obtained even the ones that were not statistically significant. [NOTE FROM THE EDITORS: we agree, all p values should be reported, including non-significant ones).	As requested, we have added all the p-values to Tables 1 and 2.	Table 1 and Table 2
*The figures mentioned in the manuscript were not present in the reviewed data available.	We are sorry that these figures were not visible for your review of the manuscript, although we were careful to closely follow the publisher's instructions. We will revise the resubmission so that you can see them.	
*References 13 (https://pubmed.ncbi.nlm.nih.gov/32065410/) and 40 (...) should be corrected.	Thank you for your comment. We have corrected both references. Koirala B, Hansen BR, Hosie A, Budhathoki C, Seal S, Beaman A, et al. Delirium point prevalence studies in inpatient settings: A systematic review and meta-analysis. Journal of clinical nursing. 2020;29(13-14):2083-92. Montes Reula L, Cañete Lairla M, Navarro López J, Pelegrín Valero C, Galindo Ortiz de Landázuri J, Marijuán Fernández P, et al. Predominant factors of institutionalization in the elderly: a comparative study between home nursing and community dwelling. Working with Older People. 2021;25(1):58-72.	References 13 and 40